# Integrative Analysis of Metabolome and Transcriptome Reveals the Mechanism of Color Formation in Yellow-Fleshed Kiwifruit

**DOI:** 10.3390/ijms24021573

**Published:** 2023-01-13

**Authors:** Yun Xiong, Junya He, Mingzhang Li, Kui Du, Hangyu Lang, Ping Gao, Yue Xie

**Affiliations:** 1College of Tropical Crops, Hainan University, Haikou 570228, China; 2Key Laboratory of Breeding and Utilization of Kiwifruit in Sichuan Province, Sichuan Provincial Academy of Natural Resource Sciences, Chengdu 610065, China; 3Key Laboratory of Bio-Resource and Eco-Environment of Ministry of Education, College of Life Sciences, Sichuan University, Chengdu 610065, China

**Keywords:** kiwifruit, flesh color, flavonoid, carotenoid, metabolites, transcriptome

## Abstract

During the development of yellow-fleshed kiwifruit (*Actinidia chinensis*), the flesh appeared light pink at the initial stage, the pink faded at the fastest growth stage, and gradually changed into green. At the maturity stage, it showed bright yellow. In order to analyze the mechanism of flesh color change at the metabolic and gene transcription level, the relationship between color and changes of metabolites and key enzyme genes was studied. In this study, five time points (20 d, 58 d, 97 d, 136 d, and 175 d) of yellow-fleshed kiwifruit were used for flavonoid metabolites detection and transcriptome, and four time points (20 d, 97 d, 136 d, and 175 d) were used for targeted detection of carotenoids. Through the analysis of the content changes of flavonoid metabolites, it was found that the accumulation of pelargonidin and cyanidin and their respective anthocyanin derivatives was related to the pink flesh of young fruit, but not to delphinidin and its derivative anthocyanins. A total of 140 flavonoid compounds were detected in the flesh, among which anthocyanin and 76% of the flavonoid compounds had the highest content at 20 d, and began to decrease significantly at 58 d until 175 d, resulting in the pale-pink fading of the flesh. At the mature stage of fruit development (175 d), the degradation of chlorophyll and the increase of carotenoids jointly led to the change of flesh color from green to yellow, in addition to chlorophyll degradation. In kiwifruit flesh, 10 carotenoids were detected, with none of them being linear carotenoids. During the whole development process of kiwifruit, the content of *β*-carotene was always higher than that of *α*-carotene. In addition, *β*-cryptoxanthin was the most-accumulated pigment in the kiwifruit at 175 d. Through transcriptome analysis of kiwifruit flesh, seven key transcription factors for flavonoid biosynthesis and ten key transcription factors for carotenoid synthesis were screened. This study was the first to analyze the effect of flavonoid accumulation on the pink color of yellow-fleshed kiwifruit. The high proportion of *β*-cryptoxanthin in yellow-fleshed kiwifruit was preliminarily found. This provides information on metabolite accumulation for further revealing the pink color of yellow-fleshed kiwifruit, and also provides a new direction for the study of carotenoid biosynthesis and regulation in yellow-fleshed kiwifruit.

## 1. Introduction

Kiwifruit cultivars (*Actinidia chinensis* and *Actinidia deliciosa*) have a bright color, rich nutrition, and a delicious taste. Anthocyanin pigments give plants pink, red, blue, and purple colors, while flavonols and flavonoid derivatives form co-coloring to enrich the color of fruits [1]. Carotenoids make fruits, flowers, and vegetables show bright colors such as yellow, orange and red [2] and can also be used as precursors of many taste-related substances, giving consumers a rich sensory experience [3]. The composition of carotenoids, along with anthocyanins and chlorophyll, accounts for the distinctive range of color found in the different kiwifruit cultivars. ‘Jinshi 1’ is a kiwifruit variety with good storage resistance and resistance to kiwifruit bacterial canker, and the flesh has a bright golden yellow. Its flesh appears pink at the early stage of development [4]. As the fruit matures, the initial pink fades rapidly and becomes green. Finally, it appears yellow at the mature stage. Therefore, it is a good material for studying the genetic mechanism of flesh color. In recent years, because of their rich functional components (carotenoids and flavonoids), kiwifruit has attracted more and more attention from consumers and researchers. This study preliminarily revealed the changing rules of flesh color of yellow-fleshed kiwifruit in terms of metabolism and genes, and the results provided theoretical reference for future breeding of colored kiwifruit.

The accumulation of anthocyanins and other flavonoids at different developmental stages is the direct factor that causes the red or purple flesh of kiwifruit [5,6]. The anthocyanin composition and content of different varieties are different [7]. Cyanidin 3-*O*-2-*O*-(*β*-xylosyl)-*β*-galactoside (Cy-xyl.gal.) and cyanidin 3-*O*-*β*-galactoside (Cy-gal.) are the two main anthocyanins in kiwifruit [7,8]. The difference in anthocyanin content derived from delphinium and cyanidin determines whether the kiwifruit is purple or red. Some key genes in the flavonoid biosynthetic pathway, such as chalcone synthase(*CHS*), chalcone isomerase(*CHI*), flavonone 3-hydroxylase(*F3H*), flavonoid-3′, 5′-Hydroxylase(*F3′5′H*), dihydroflavonol 4-reductase(*DFR*), anthocyanin synthase(*ANS*), etc., showed a great influence on flower color modification [9]. In terms of gene regulation, R2R3-MYB, basic helix-loop-helix (bHLH) and WD40 protein-binding transcription factor complexes are considered to be important transcription factors regulating genes with flavonoid structure [10].

Previous studies have shown that the chlorophyll and carotenoid content of yellow-fleshed kiwifruit of different genotypes has different effects on the color change, and the carotenoid content and composition of yellow-fleshed kiwifruit at different developmental stages are also different. *A. deliciosa* has green flesh when ripe, whereas *A. chinensis* often has yellow flesh when ripe. The flesh of *A. deliciosa* fruit retains its chlorophyll during ripening. In those *A. chinensis* fruits that turn yellow at maturity, the color change is due to the disappearance of chlorophyll, exposing the yellow carotenoids that are already present, rather than an increase in carotenoid synthesis [11]. In *A. deliciosa*, a high concentration of chlorophyll remains in the flesh of mature green kiwifruit. In *A. chinensis*, chlorophyll degrades into colorless catabolites when the fruit matures, leaving visible yellow carotenoids. The chlorophyll degradation of kiwifruit is differently regulated, and golden kiwifruit transcribes more degradation genes, which leads to chlorophyll degradation in golden kiwifruit earlier and longer than green kiwifruit [12]. The carotenoid content of *A. macrosperma* and its hybrid with *A. melanandra* is higher at 20 d [13]. As the fruit ripens, carotenoid content declines again, and increases as the fruit softens, directly affecting the flesh color. Lutein and *β*-carotene are the most abundant carotenoids found during fruit development. *β*-carotene content increases rapidly during fruit ripening [13]. GGPP is a common precursor for the biosynthesis of carotenoids [14]. Two molecules of geraniyl geraniyl pyrophosphate (GGPP) were catalyzed by the phytoene synthase(PSY) to form phytene. The colorless phytene then produces *ζ*-carotene and lycopene, which requires phytene desaturase (PDS) and zeta-carotene desaturase (ZDS). The carotenoid pathway branches when lycopene cyclizes, and lycopene cyclase acts on lycopene cyclization to produce *α*-carotene and *β*-carotene. LCY-*ε* and LCY-*β* are cyclases that produce lutein or *β*-carotene, and their ratio determines the color of the fruit [13,15]. In the kiwifruit, the expression of *ZDS* and *LCY-β* genes increased, and the expression of *LCY-ε* and *CRHb* genes was down-regulated. The changes in these genes together control the accumulation of *β-carotene*. The differences in gene expression between different genotypes may be related to the differences in the regulation of these genes, which requires further study [13].

At present, most metabolic regularity and molecular regulation mechanisms for flavonoids have focused on the green-fleshed kiwifruit, and the accumulation characteristics of flavonoids and their gene regulation mechanisms in yellow-fleshed kiwifruit are still unclear. In the study of carotenoid accumulation in kiwifruit, although the functions of important genes in the carotenoid metabolism pathway have been proven, the correlation between the pigment content and the dynamic changes in the flesh color is still not perfect. More importantly, as the variety has the darkest yellow flesh, the composition of carotenoids in the flesh and the molecular regulation mechanism of accumulation are still unclear. Therefore, ‘Jinshi 1’ (*A. chinensis*) was selected in this experiment, and ultra-high performance liquid chromatography-tandem mass spectrometry (UPLC-ESI-MS/MS) technology was used to detect flavonoids and carotenoids in the flesh at different developmental stages. In addition, this experiment also performed transcriptome sequencing of the flesh, and found structural genes and transcription factor genes related to metabolites, and analyzed the impact of key genes on metabolites. Transcription factor genes related to metabolite regulation were screened out through correlation analysis and the weighted gene co-expression network analysis (WGCNA) method. This experiment provided new insights into the transition of different colors during fruit development and also screened key candidate transcription factor genes for the later exploration of the molecular mechanism.

## 2. Results

### 2.1. Transcriptome Profiling

After 20 d, the fruit was round and oval, with white core, green skin and pale-pink flesh. Fruit experienced the fastest growth stage (cell division) from 20 d to 58 d and then grew slowly until fruit expansion (58 d to 97 d). The young fruit was hard and green. At stage 97 d, fruit growth had slowed considerably, and the seeds had begun to change color, first from white to brown, then finally to black. The flesh began to turn color. The flesh color began to decline until stable. Over 95% of the seeds had turned black at 136 d. Fruit is ripe for commercial picking at 175 d. According to the chroma value of flesh [16], five key development periods were selected for the experiment, corresponding to the fruit tender time points (T1, 20 d), early color transition time points (T2, 58 d), color change time points (T3, 97 d), late color transition time points (T4, 136 d), and fruit maturity stage (T5, 175 d), respectively (Figure 1A). RNA-seq was performed to analyze the level of transcription in flesh during the fruit development process, using equal amounts of RNA for five development stages. In total, we obtained 240.66 Gb of high-quality reads (Q30 > 85%) without an adaptor (Appendix A).

### 2.2. Identification of Flavonoids in the Fruits of Five Development Stages

The flavonoids in the flesh of five periods (20 d, 58 d, 97 d, 136 d, and 175 d) were thoroughly qualitatively and quantitatively analyzed in order to analyze the impact of flavonoids and anthocyanins on the color appearance of fruits. The relative content of metabolites was measured using the multiple reaction monitoring (MRM) model and is listed in Appendix A. A total of 140 flavonoid compounds were detected, including 8 isoflavones, 12 flavanones, 14 flavonoids, 27 flavonols, 61 flavones, 18 anthocyanins, and proanthocyanidins. It can be seen from the heatmap that 76% of the flavonoids have the highest content in the 20 d, while the relative content in the later fruit development time points (136 d, 175 d) is relatively low, and a few substances have a high content in 97 d or 136 d (Appendix A). In ‘Jinshi 1’, cyanidin, pelargonidin, peonidin chloride and four anthocyanins derived from them (cyanidin-3-*O*-galactoside chloride, cyanidin-3-*O*-glucoside, pelargonidin-3-*O*-glucoside and malvidin-3-*O*-glucoside) were detected, while delphinidin and its derived anthocyanins were not (Figure 1B). Pelargonium was only detected for 20 d, and the other periods were below the detection limit. The cyanidin and its derived anthocyanins were highest in the 20 d and began to decline in the 58 d, and were the lowest in the 136 d and 175 d (Figure 1B).

### 2.3. Expression of Flavonoid Synthesis-Related Genes in Five Development Stages

Anthocyanins pass through phenylpropane and flavonoid metabolic pathways, respectively. Therefore, the accumulation of anthocyanins is closely related to the enzymatic activity of metabolic pathways. In the upstream of the metabolic pathway (Figure 2, Appendix A), 4 *PAL* and *C4H* have the highest expression levels in 20 d, and three *4CL* have the highest expression levels in 20 d or 58 d. In addition, the content of the substrate phenylalanine also dropped to the lowest level during 175 d, which may not provide sufficient raw materials for downstream synthesis of flavonoids. The transcription level of *F3′5′H* is below the detection limit. The expression of the 5 *FLS* genes was higher at 20 d and 58 d and decreased at 175 d. *DFR1* and *DFR3* were also highest in the first two periods. *LDOX* was highest at 20 d and lowest at 175 d. To validate the RNA-Seq results, we chose 10 flavonoid biosynthetic pathway genes (Appendix A) and used qRT-PCR to examine their expression levels after 20 d, 58 d, and 97 d. Among the ten genes, except for *4CL*, *F3H*, and *ANR*, which were higher than 20 d in the other two periods, the expression levels of other genes were lower than 20 d. The latter two periods were too low to detect.

### 2.4. Candidate Transcription Factor Genes Related to Flavonoid Synthesis

A transcription factor complex containing R2R3-MYB, basic-helix-loop-helix (bHLH) and WD40 proteins (named MBW complexes) has been reported to regulate the activity of structural genes [17]. Genes with the same expression pattern may have similar biological functions. In order to find transcription factors related to flavonoid synthesis, differentially expressed genes were divided into six gene types according to the different expression patterns of FPKM values in different periods (Figure 3A). Class 3 is the set that contains the largest number of genes (5847). The expression level of genes is the highest in 20 d, and after a significant decrease in 58 d, and maintains low expression levels in the later time points. In this grouping, besides being enriched in the flavonoid pathway, the genes in Class 3 are also enriched in the plant hormone signal transduction pathway and the starch and sugar metabolism pathway (Figure 3B). The transcription factor genes in this gene cluster were classified and counted, and the main transcription factor families were found to be AP2/ERF-ERF (14.6%), C2H2 (9.5%), bHLH (9.2%), NAC (7.8%), WRKY (7.8%), MYB-related (6.6%), bZIP (4.6%) and MYB (3.9%) (Figure 3C). In order to further search for candidate transcription factors in Class 3, we calculated the correlation coefficients between structural genes in the flavonoid metabolism pathway and transcription factor family (MYB, bHLH) genes, filtered the relationship pairs below 0.9 and obtained 327 relationship pairs (Appendix A). In the co-expression network of structural genes and transcription factors, the greater the degree of a gene, the more genes that have an interaction relationship with it. In this experiment, seven TFs were screened, all with degrees above nine. The TFs as a hub gene may play an important regulatory role in the network. They are *bHLH29* (*Achn335591*), *bHLH18* (*Achn021041*), *MYB7* (*Achn102731*), *bHLH31* (*Achn039371*), *MYB5* (*Achn324811*), *bHLH25* (*Achn087151*) and *bHLH28* (*Achn348771*), and these transcription factors are positively correlated with other genes (Appendix A).

### 2.5. Expression of Carotenoid Synthesis Genes in Four Development Stages

The total amount of carotenoids (Figure 4A) reached 49.22 μg/g at 20 d, and then decreased at 97 d and 136 d to 33.80 and 40.54 μg/g, respectively. The content quickly reached the highest at 175 d (73.25 μg/g). The content of total chlorophyll (Figure 4B) was the highest at 97 d (10 μg/g), approached the detection limit at 136 d, and remained low (7.3 μg/g) at 175 days. In short, carotenoids increase in the later stage of fruit development, while the content of chlorophyll decreases. 10 carotenoids (*α*-carotene, *α*-cryptoxanthin, *β*-carotene, apocarotenal, *β*-cryptoxanthin, lutein, neoxanthin, zeaxanthin, violaxanthin and antheraxanthin) in the pulp of ‘Jinshi 1’ kiwifruit were identified (Figure 4C, Appendix A). In this experiment, the total chlorophyll decreased significantly during the color transition period (136 d) and the late color transition period (175 d), while the chlorophyll degradation-related genes, *PPH3*, *PAO1*, *PAO2* and *SGR1*, all showed an increasing trend (Appendix A). Among the synthesis-related genes, *GluTR1*, *LHCB1* and *LHCB2* are expressed at a lower level in the late stage of color transformation. *CAO1* has the highest expression level in the 20 d, and the lowest level in the 97 d, while *CBR1*, *CLH1* and *CLH2* have the lowest expression level in the 175 d, which is inconsistent with the expression trends of other chlorophyll synthesis-related genes. *PSY1*, *PSY2* and the total carotenoid content have a similar change trend, while *PSY1* and *PSY2* have the lowest expression in the 97 d. In the 20 d and 175 d, they are relatively high, and the expression is the highest in the 136 d. The expression levels of *PDS*, *ZDS* and *CrtISO* genes were the highest at 136 d, which was inconsistent with the total change of carotenoids (Appendix A). Lycopene cyclization is a major branch point in carotenoid metabolism. On the one hand, the upstream lycopene forms α-carotene under the action of lycopene *ε*-cyclase (*LCYE*), and on the other hand, Lycopene *β*-cyclase (*LCYB*) forms *β*-carotene. In this experiment, it was detected that the two *LCYE1* and *LCYE2* genes were the highest in the 20 d, and their expression levels decreased in the later stage of fruit development. At the same time, the *LCYB2* and *LCYB3* genes were lower in the first 20 d and increased in the 136 d and 175 d (Figure 4D). In addition, the expression of the *LCYB* gene increased significantly during 175 d in qRT-PCR experiment (Appendix A). The enzymatic reaction efficiency of the enzyme LCY-*β* at the fruit 175 d stage may be higher than that of the cyclase LCY-*ε*, which promotes the conversion of lycopene to the direction of *β*-cryptoxanthin.

### 2.6. Co-Expression Network Analysis Identified Carotenoid-Related DEGs

In order to screen the regulatory genes related to carotenoid metabolism, the genes in the 20 d, 97 d, 136 d, and 175 d were analyzed by WGCNA. According to the expression pattern, they can be divided into 11 modules (Figure 5A and Appendix A), namely “black”, “blue”, “brown”, “green”, “grey”, “magenta”, “pink”, “purple”, “red”, “turquoise” and “yellow” gene matrices of different modules; these matrices are the same as C01 (*α*- Carotene), C02 (*β*-carotene), C04 (*ε*-carotene), C05 (lutein), C07 (antheraxanthin), C08 (neoxanthin), C09 (zeaxanthin), C10 (*β*-cryptoxanthin), C12 (apocarotenal), and C17 (*α*-cryptoxanthin). According to the correlation and *p*-value, the two modules “blue” and “turquoise” have the most significant correlation with carotenoids, and the “blue” module has a very significant positive correlation with *β*-cryptoxanthin and *α*-cryptoxanthin. Correlation, the cor value and *p*-value are 0.99 (4 × 10^−9^) and 0.98 (1 × 10^−8^), respectively. In addition, this module is also negatively correlated with lutein, antheraxanthin, and zeaxanthin. The cor value and p value are, respectively −0.95 (4 × 10^−6^), −0.96 (7 × 10^−7^) and −0.92 (3 × 10^−5^). The “turquoise” module has a significant positive correlation with *α*-carotene, *β*-carotene, lutein, neoxanthin, zeaxanthin, and *β*-apocarotene. The correlations are 0.87 (2 × 10^−4^) and 0.95, respectively. (3 × 10^−6^), 0.81 (0.001), 0.97 (9 × 10^−8^), 0.87 (2 × 10^−4^), 0.98 (2 × 10^−8^).

KEGG enrichment analysis was performed on the genes of the “blue” module, and the results (Figure 5B) showed that the main enriched items of genes were terpenoid backbone biosynthesis, riboflavin metabolism, pyruvate metabolism, carbon metabolism, citrate cycle, proteasome, protein export, protein processing in endoplasmic reticulum, ribosome biogenesis in eukaryotes, nicotinate and nicotinamide metabolism, glycolysis/gluconeogenesis and fatty acid metabolism. Perform KEGG enrichment analysis on the “turquoise” module, and the results (Appendix A) show that the significantly enriched items include sphingolipid metabolism, plant hormone signal transduction, and other polysaccharides. Glycan degradation, glycosaminoglycan degradation. *β*-cryptoxanthin accounts for 92.3% of the total carotenoids at 175 d, which is the main carotenoid accumulated in fruits. The correlation between the genes in the “blue” module and *β*-cryptoxanthin reaches 0.99, and it is significant when it reaches 4 × 10^−9^. In addition, the genes of the “blue” module are also enriched in the biosynthesis of terpenoids. Carotenoids are fat-soluble and are a group of tetraterpenoids found in vegetables and fruits. In addition, the “blue” module was selected as the key target module for this experiment. A large number of transcription factor families are enriched in the “blue” module, such as NAC, bHLH, B3-ARF, AP2/ERF-ERF, Bzip, etc. There are a total of 182 transcription factors (Appendix A), and transcription factors in plants. The database (Plant Transcription Factor Database, http://plantregmap.cbi.pku.edu.cn/network.php (accessed on 18 May 2018) screened out the predicted transcription factors that regulate structural genes in the carotenoid metabolic pathway (Appendix A). The transcription factors enriched in the “blue” module are crossed with the predicted transcription factors in the database, and 10 candidate transcription factors related to carotenoid metabolism are obtained (Appendix A).

## 3. Discussion

### 3.1. Effects of Anthocyanins and Flavonoids on Flesh Color at the Young Fruit Stage

The flesh color of the whole development period changes dynamically with development. The difference in anthocyanin content derived from cyanidin and delphinidin determines whether kiwifruit appears red or purple [5,6]. Derivatives of delphinidin were only found in *A. melanandra* and *A. arguta* [18]. Anthocyanins derived from cyanidin were found in both *A. arguta* and *A. chinensis*. *A. arguta* mainly contained cyanidin 3-*O*-galactoside and cyanidin 3-*O*-glucoside, and *A. chinensis* mainly contained cyanidin 3-*O*-xylo-(1-2)-galactoside. It also contains a small amount of cyanidin 3-*O*-galactoside [7]. Colorless flavonoids are the most common copigments in higher plants. Copigments themselves have no direct effect on color, but enhance the color effect of anthocyanins by affecting the combination of anthocyanins and their derivatives. In addition, when the anthocyanin components are the same but the copigments are different, the color of the petals will be different [19,20]. In ‘Jinshi 1’, cyanidin, pelargonidin, peoniflorin, and their derivatives were detected, but delphinidin and its derivatives were not detected. The contents of cyanidin, pelargonidin and delphinidin in red-fleshed kiwifrui were significantly higher [21]. Therefore, the light-red color of yellow-fleshed kiwifruit is mainly attributed to pelargonidin and cyanidin and their derived anthocyanins, and the chromogenic effect of delphinidin is not reflected. Anthocyanins and their derivatives, 76% flavonoids were the highest in the 20 d of the fruit, and decreased significantly at 58 d. The total amount of flavonoids in other yellow-fleshed kiwifruit varieties (‘Jinfeng’, ‘Jinkui’) also gradually decreased [5]. These flavonoids and anthocyanins may be rapidly transformed into other substances to support other secondary metabolic activities after 20 d, and the co-pigmentation effect of flavonoids on anthocyanins is also weakened with the decrease of flavonoids. The types and quantities of flavonoids are different in different species, different tissues, and developmental stages. Isoflavones are mainly present in legumes [22], while flavonols and proanthocyanidins are present in *Arabidopsis* seeds [23]. A total of 139 flavonoid metabolites were identified in kiwifruit, belonging to six different categories of flavonoids, including flavonols, anthocyanins and proanthocyanidins, flavanones and isoflavones.

Flavonoids are syntheised through the phenylpropanoid pathway [24]. The structural genes *PAL* and *4CL* encode the corresponding enzymes, which convert phenylalanine into 4-coumarin coenzyme A and then begin to enter the synthesis of flavonoids [25]. In ‘Jinshi 1’, the expression levels of four *PAL* genes were the highest at 20 d, and the expression levels of several *4CL* genes were inconsistent. Therefore, there may be a mechanism targeting the *PAL* gene, which plays an important role in the synthesis of flavonoids in the downstream pathway, resulting in a significant decrease in the content of most flavonoids (76%) in the late stage of fruit development. In the upstream of the flavonoid metabolic pathway, the naringin ligand chalcone produces dihydroflavonol under the action of isomerase CHI and hydroxylase F3H [26]. It can be catalysed by enzymes F3′H and F3′5′H to produce different types of colored anthocyanins to present purple or red [27]. In *A. chinensis*, the enzyme activity of F3′H was detected, while the enzyme activity of F3′5′H was not detected, and only *F3′H* gene was expressed, while the *F3′5′H* gene was not expressed. In contrast to the results for *A. chinensis*, the activity of F3′5′H was also detected in *A. arguta* besides *F3′H* [28]. In ‘Jinshi 1’, the transcription information of *F3′5′H* was not detected, which was similar to the results in kiwifruit, indicating that the *F3′H* gene in the upstream pathway may be more important than *F3′5′H* for the synthesis of anthocyanin compounds in ‘Jinshi 1’fruit. In the downstream of flavonoid biosynthesis pathway, enzyme DFR and LAR share anthocyanin and flavanone biosynthesis pathway. Dihydroquercetin, dihydrokaempferol and dihydromyricetin have different substrate biases for DFR. In addition, each LAR has a specific c-terminal domain, which also has different substrate specificities [29]. In ‘Jinshi 1’, cyanidin and pelargonidin and their derivatives are mainly related to the pink phenotype. Therefore, in ‘Jinshi 1’, enzymes DFR and LAR are more inclined to produce cyanidin and pelargonidin with dihydroquercetin and dihydrokaempferol as substrates, rather than synthesize delphinidin. As a key transcription factor in all anthocyanin biosynthesis pathways, MYBs play a key role in regulating the color formation of fruits and flowers [17]. CHS is positively regulated by *MYB4* and *MYB5* expression, and strawberry *FcMYB1* has a switch effect on anthocyanin and flavonoid accumulation [30]. The deletion of the MYB cis-element in the *CHS* promoter resulted in the formation of white fruit. In this study, two MYB genes, *MYB7* (*Achn102731*) and *MYB5* (*Achn32481*), were screened out as hub genes based on connectivity, which may play an important regulatory role in the network. These transcription factors were positively correlated with other genes, indicating that MYB is a key regulator of anthocyanin and flavonoid biosynthesis pathways in kiwifruit.

### 3.2. Effects of Carotenoids and Chlorophyll on Flesh Color at Ripening

The difference in composition and content of chlorophyll and carotenoids in kiwifruit resulted in green and yellow flesh. When ‘Hayward’ matured, both the outer and inner flesh retained chlorophyll, while in yellow-fleshed kiwifruit ‘Hort16A’and ‘Jinfeng’, chlorophyll degraded during maturation, exposing the original carotenoids in the fruit, resulting in the color changing from green to yellow [18]. In ‘Jinshi 1’, the total chlorophyll content began to decrease significantly at 136 d, and the content at 175 d was still lower than that at 20 d and 97 d. The total carotenoid content reached 49.22 μg/g at 20 d, decreased to 33.80 μg/g at 58 d, and then increased significantly to 73.25 μg/g at the late coloration stage, which was significantly higher than the other three stages. The degradation of chlorophyll and the accumulation of carotenoids caused the flesh color of ‘Jinshi 1’ to change from green to yellow, which was different from other yellow-fleshed kiwifruit. Although they belonged to yellow-fleshed kiwifruit, the difference of genetic background may lead to different accumulation patterns of pigments.

Among the carotenoid components of *A. chinensis* and *A. deliciosa*, lutein and carotene are the main components. Lutein includes 9′-*cis*-neoxanthin, violaxanthin, antheraxanthin, zeaxanthin and *β*-cryptoxanthin. Carotene is mainly *β*-carotene and *α*-carotene [7]. In ‘Jinshi 1’, a total of 10 carotenoids were detected, while phytoene and phytoene were not detected, which was consistent with the detection results of Chinese kiwifruit [13]. Therefore, the lower detection limit of linear carotene may also be due to effective enzymatic reactions or upstream limiting factors. In the downstream branch pathway, the content of *β*-carotene reached the highest at 20 d, and the content of each period was higher than that of *α*-carotene. The content of *β*-carotene was always higher than that of *α*-carotene in the three progeny individuals of the cross breeding of large seed kiwifruit and black stamen kiwifruit. Therefore, compared with α-carotene, *β*-carotene is the main accumulated carotenoid in ‘Jinshi 1’. In the branch pathway of carotenoid metabolism, it was also found that *β*-cryptoxanthin gradually increased with fruit development, reaching the highest content in the 175 d, accounting for 92.3% of the total carotenoid content. It was the most accumulated pigment in the fruit-picking period, and the content accumulated fastest at 136 d. Lycopene cyclization is a major branch point in carotenoid metabolism. On the one hand, the upstream lycopene forms α-carotene under the action of lycopene *ε*-cyclase (LCY-*ε*), and on the other hand, it forms β-carotene under the action of lycopene *β*-cyclase (LCY-*β*) [31]. This experiment detected that the two *LCYE1* and *LCYE2* genes were the highest at the 20 d stage, and the expression level decreased in the later stage of fruit development. At the same time, the expression levels of *LCYB2* and *LCYB3* genes were low at the 20 d stage, and the expression levels increased at 136 d and 175 d stages. Therefore, the enzymatic reaction efficiency of the cyclase LCY-*β* at the 175 d stage of the fruit may be higher than that of the cyclase LCY-*ε*, which leads to the conversion of lycopene mainly to the direction of *β*-cryptoxanthin, and ultimately leads to the accumulation of *β*-cryptoxanthin. In kiwifruit, the expression levels of structural genes such as *PSY*, *ZDS* and *LCYB* were consistent with the increasing trend of carotenoid content [13]. In ‘Jinshi 1’, the changing trend of *PSY1* and *PSY2* was similar to that of total carotenoid content, while the expression of *PSY1* and *PSY2* was the lowest at 97 d, relatively higher at 20 d and 175 d, and the highest at 136 d. The expression levels of *PDS*, *ZDS* and *CrtISO* genes were the highest at 136 d, which was inconsistent with the change in total carotenoid content. Therefore, the structural gene *PSY* in ‘Jinshi 1’ may play an important role in carotenoid accumulation. Carotenoid synthesis in fruit is regulated not only by structural genes but also by plant hormones. The ripening of climacteric fruits is triggered by the rapid synthesis of ethylene. The transcription factor *SlAP2a* in tomato is induced and negatively regulated during the ripening of tomato fruits. The inhibition of *SlAP2a* expression leads to excessive ethylene production in fruits and leads to ripening [32]. It also changes carotenoid accumulation by changing carotenoid pathway flux. Three transcription factor genes (*Achn266591*, *Achn310981*, *Achn187281*) were annotated as ethylene responsive transcription factors in the protein database.

Chlorophyll metabolism was divided into three stages: chlorophyll synthesis, chlorophyll cycle and chlorophyll degradation. Among them, the relative expression levels of chlorophyll synthesis-related genes *RBCS* and *CAO* were decreased in yellow meat ‘Hort16A’, resulting in decreased chlorophyll biosynthesis. In yellow-fleshed (*A. chinensis*) and green-fleshed (*A. deliciosa*), the chlorophyll degradation gene *PAO* was expressed, and the expression level was higher in yellow flesh. The gene can catalyze the conversion of pheophorbide A into chlorophyll decomposition, thereby reducing the chlorophyll content in the flesh. The stay-green gene (*SGR*) can convert chlorophyll into free chlorophyll and enter the degradation pathway. It is a key gene for chlorophyll degradation, and the expression level of this gene in yellow-fleshed kiwifruit is higher than that in green-fleshed kiwifruit [12]. In ‘Jinshi 1’, the total chlorophyll content decreased significantly at 136 d and 175 d, while the chlorophyll degradation related genes *PPH3*, *PAO1*, *PAO2* and *SGR1* showed an increasing trend. The expression levels of *GluTR1*, *LHCB1* and *LHCB2* in chlorophyll synthesis related genes were lower in the late stage of color conversion. The expression level of *CAO1* gene was the highest at 20 d and the lowest at 97 d, while the expression levels of *CBR1*, *CLH1* and *CLH1* were the highest at 175 d, which was inconsistent with the expression trends of other chlorophyll synthesis-related genes. Therefore, the increased expression of chlorophyll degradation-related genes and the decreased expression of chlorophyll synthesis-related genes may lead to the decrease of chlorophyll content.

## 4. Materials and Methods

### 4.1. Plant Material and Sample Preparation

‘Jinshi 1’ was planted in the experimental field of kiwifruit in Shifang City, Sichuan province, China, in spring 2018. Harvesting of kiwifruit occurred at nine different stages after full bloom (Figure 1A). The sampling area in the kiwifruit is the outer pericarp (Appendix A). All samples were immediately frozen in liquid nitrogen and stored at −80 °C. The flesh was vacuum freeze-dried and crushed (30 Hz, 1.5 min) with a mixer mill (MM 400, Retsch) to a powder form, and 100 mg of the powder was taken and dissolved in 1.0 mL methanol of the extract. The samples dissolved with methanol were stored at 4 °C for 12 h, centrifuged at 10 k rpm for 10 min, the supernatant was aspirated, filtered and stored in the sample bottle. The experimental design was completely randomized and consisted of three biological replicates for each of the treatments.

### 4.2. Metabolite Profiling Analysis, and Representative Flavonoid Compound Measurement

Extraction and sample preparation were performed according to a protocol described previously [33]. The data acquisition instrument system mainly includes Ultra Performance Liquid Chromatography (Shim-pack UFLC, Shimadzu CBM30A) and Tandem Mass Spectrometry (MS/MS) (Applied Biosystems 6500 QTRAP). The UPLC analytical conditions were as follows: column, Waters ACQUITY UPLC HSS T3 C18 (1.8 μm, 2.1 mm × 100 mm); solvent system, water (0.04% acetic acid): acetonitrile (0.04% acetic acid); gradient program, 100:0 V/V at 0 min, 5:95 V/V at 11.0 min, 5:95 V/V at 12.0 min, 95:5 V/V at 12.1 min, 95:5 V/V at 15.0 min; flow rate, 0.40 mL/min; temperature, 40 °C; injection volume: 2 μL. The MS conditions mainly include ESI, 500 °C; CUR, 25 psi. Metabolite quantification was performed using a scheduled multiple reaction monitoring (MRM) method [34].

### 4.3. Carotenoid Identification and Quantification

Kiwifruit flesh (100 mg weight) was frozen in liquid nitrogen, ground into powder, and extracted with n-hexane: acetone: ethanol (2:1:1, V/V/V). The extract was vortexed (30 s), ultrasound-assisted extraction was carried out for 20 min at 25 °C, and centrifuged (12 k rpm, 5 min). After centrifugation, supernatant was taken, and the above centrifugation steps were repeated to obtain supernatant. After the supernatant was collected twice, it was evaporated under nitrogen gas to dry and recombined in 75% methanol (V/V). The sample extracts were analyzed using an LC-ESI-MS/MS system (UHPLC, ExionLC™ AD; MS, Applied Biosystems 6500 Triple Quadrupole). The analytical conditions were as follows: HPLC: column, YMC C30 (3 µm, 2 mm × 100 mm); solvent system, acetonitrile: methanol (3:1, V/V) (0.01% BHT): methyl tert-butyl ether (0.01% BHT); gradient program, 85:5 V/V at 0min, 75:25 V/V at 2 min, 40:60 V/V at 2.5 min, 5:95 V/V at 3min, 5:95 V/V at 4.0 min, 85:15 V/V at 4.1 min, 85:15 V/V at 6 min; flow rate, 0.8 mL/min; temperature, 28 °C; injection volume: 5 μL. Methanol, acetonitrile, ethanol and acetone used in the test were produced from Merck, and various carotenoid (*α*-carotene, *β*-carotene, *γ*-carotene, *ε*-carotene, lutein, violaxanthin, antheraxanthin, neoxanthin, zeaxanthin, *β*-cryptoxanthin, lycopene, phytofluene, *(E/Z)*-phytoene, astaxanthin, capsanthin, apocarotenal and *α*-cryptoxanthin, capsorubin) standards were produced from Sigma. The effluent was alternatively connected to an ESI-triple quadrupole-linear ion trap (QTRAP)-MS. API 6500 Q TRAP LC/MS/MS System, equipped with an APCI Turbo Ion-Spray interface and operating in a positive ion mode. The APCI source operation parameters were as follows: ion source, turbo spray; source temperature 350 °C; curtain gas (CUR) was set at 25.0 psi; the collision gas (CAD) was medium. Qualitative analysis of MS data was carried out by using the plant carotenoid database in Metware database [34] (MWDB, Wuhan, China). The LC-MS/MS data of carotenoids were analyzed using Analyst 1.6.3 software (AB Sciex, Framingham, MA, USA), and various MRM changes and integrations were automatically identified using default parameters. By preparing standard carotenoid solutions of different concentrations, the mass spectrum peak intensity data of the corresponding quantitative signal of the standard solution of each concentration was obtained. The standard curves of different carotenoids were plotted with the standard concentration (μg/mL) as the horizontal coordinate and the peak area of the mass spectrum peak as the vertical coordinate (Appendix A).

### 4.4. RNA Isolation and Transcriptome Sequencing

Total RNA was extracted from flesh tissues of the outer pericarp using a modified Trizol protocol (Invitrogen Corp., Carlsbad, CA, USA). Briefly, collected tissue from each sample was finely ground in liquid nitrogen. For each 100 mg of tissue, 1 mL of chilled Trizol was added. Samples were centrifuged at 10 k rpm for 5 min at 4 °C. Supernatant was discarded and 200 μL of chilled chloroform was added and vortexed at high speed for 20 s. Samples were centrifuged again at 10 k rpm for 15 min at 4 °C. The aqueous phase was mixed with 0.89 isopropanol and left at 25 °C for 10 min, followed by centrifugation at 10 k rpm at for 10 min at 4 °C. The supernatant was discarded, and the pellet was washed. Samples were cleaned using the RNeasy Plant Mini Kit (Qiagen, Hilden, Germany) with 75% ethanol. Pellet was air-dried for 10 min at 25 °C and resuspended in 20 μL of nuclease-free water followed by incubation at 50 °C for 12 min. RNA concentration and purity were determined by a Qubit^®^ RNA Assay Kit (BR) in a Qubit 2.0 fluorometer (Life Technologies, Carlsbad, CA, USA) and a NanoPhotometer^®^ spectrophotometer (Implen, Westlake Village, CA, USA), respectively. RNA integrity was assessed using the RNA Nano6000 Assay Kit on a Bioanalyzer 2100 system (Agilent Technologies, Santa Clara, CA, USA). As an input material for the RNA sample preparations, 3 μg of RNA per sample was used as an input. Sequencing libraries were generated using a NEBNext^®^ Ultra™ RNA Library Prep Kit for Illumina^®^ (NEB, Ipswich, MA, USA) following default parameters, and index codes were added to attribute sequences to each sample. The library preparations were sequenced on an Illumina Hiseq platform (Illumina, San Diego CA, USA), and 125 bp/150 bp paired-end reads were generated. The RNAseq experiment included three biological replicates per treatment. All genes and annotation information obtained by sequencing are shown in Appendix A.

### 4.5. Statistical Analysis

Gene expression levels were measured using StringTie software. DESeq2 [35] was used for the analysis of differentially expressed genes (DEGs), and a multiple hypothesis test correction was applied to the hypothesis test probability (P value) by the Benjamini–Hochberg method. The screening condition for DEGs was |log2(fold change)| ≥ 1, and false discovery rate (FDR) < 0.05. The identified DEGs were further subjected to enrichment analysis through KEGG pathway analysis. Cluster analysis and heatmap were performed by pheatmap program package of R language. Correlation analysis through the R language packages, metabolites column, and line chart using the R language plotly website (https://chart-studio.plotly.com/create (accessed on 23 January 2018)) online mapping. Co-expression networks were presented to find the relations among genes [36]. Networks were built according to the normalized expression values of genes selected from significant GO terms. The Pearson correlation was calculated for each pair of genes. Significant correlation pairs (FDR < 0.05) were chosen to construct the network. Within the network, the relative importance of each gene is determined by its degree of centrality. Degree centrality is defined as the link numbers at one node. The hub genes in the network were identified according to the topological coefficient of each node with a degree >9 or betweenness centrality >0.05. In this study, WGCNA package in R package was used to construct gene co-expression network. Genes with FPKM gene expression greater than 1 were introduced into WGCNA. All genes were divided into different modules according to their expression patterns, which were used to construct scale-free networks. In order to ensure the construction of scale-free network, the soft threshold is used as the selection criterion. Pearson correlation coefficient and *p*-value between characteristic genes and characteristic metabolites of each gene module were used for calculation. Among the most relevant gene modules, the target genes with the highest connectivity were screened, and the “dot” and “line” files of the target genes were imported into cytoscape 3.9.0. In the gene expression trend analysis, the FPKM of the gene was increased by 1 and then standardized and centralized, and then the K-means cluster analysis was performed.

### 4.6. qRT-PCR Analysis

The total RNA was extracted from the flesh of four time points (20 d, 97 d, 136 d, 175 d) according to the Unlq-10 Column Trizol Total RNA Extraction Kit (Sangon Biotech, Shanghai, China) instructions. The first strand of cDNA was synthesized using reverse transcriptase (GeneRuler DNA Ladder Mix, Shanghai, China). Twelve genes related to flavonoid and carotenoid metabolic pathways were selected for qRT-PCR analysis, and the actin gene was used as a reference gene to correct gene expression. All primers used in this study are listed in Appendix A. Reactions were carried out on a StepOne Plus Real-Time PCR detection system (ABI, Foster, CA, USA) using a qPCR Master Mix (SYBR Green) kit (High Rox, B639273, BBI). The run conditions were 2 min of initial denaturation at 95 °C, 95 °C for 5 s, 58 °C for 10 s and 72 °C for 20 s (40 cycles). Three replicates were performed for each sample. Quantitative data was analyzed using the 2^−ΔΔCT^ method.

## 5. Conclusions

During the fruit development of ‘Jinshi 1’ yellow-fleshed kiwifruit, pelargonidin, cyanidin and their respective anthocyanin derivatives led to pink flesh at the young fruit stage. The content of anthocyanin and 76% of the flavonoid compounds was the highest at 20 d, and decreased significantly at 58 d to other metabolites to support other secondary metabolic activities, so that the pale-pink of the flesh disappeared. In the late stage of fruit development, the degradation of chlorophyll and the accumulation of carotenoids led to the color of the flesh to change from green to yellow, rather than the unilateral reduction of chlorophyll. A total of 10 carotenoids were determined in the flesh, but linear carotenoids were not detected. Among all carotenoids, the content of *β*-carotene was always higher than that of *α*-carotene. In addition, *β*-cryptoxanthin gradually increased with fruit development and was the most-accumulated pigment in the fruit during the picking time points (175 d). Through transcriptome analysis of kiwifruit, 7 key transcription factors for flavonoid biosynthesis and 10 key transcription factors for carotenoid synthesis were screened.

## Figures and Tables

**Figure 1 ijms-24-01573-f001:**
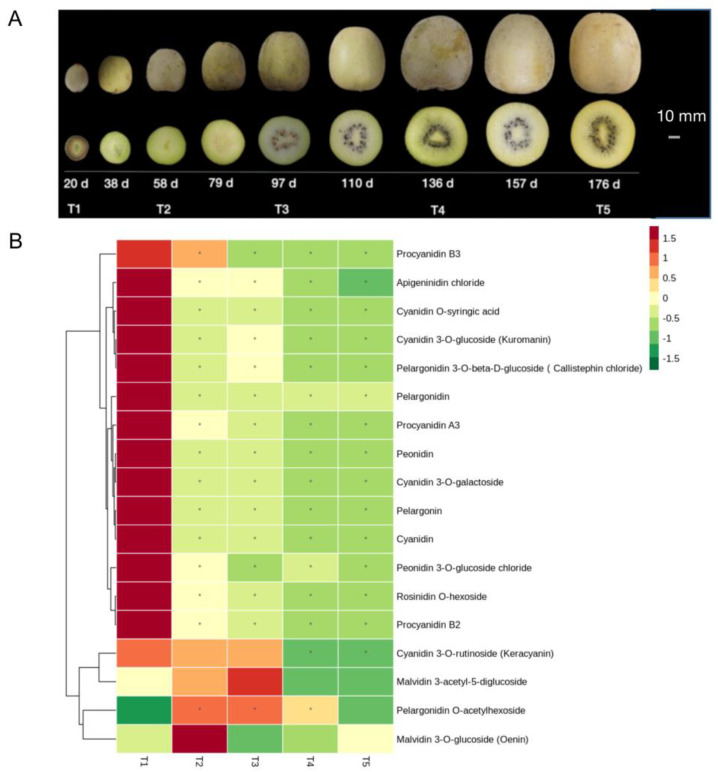
(**A**): Morphological changes in nine developmental stages. (**B**): Cluster heat map of relative content of anthocyanin and procyanidins. Each line in the heatmap represents a metabolite. The deeper the red color, the higher the level of that metabolite in developing fruits. Similarly, the deeper the green color, the lower the level of that metabolite in developing fruits. The five cells from left to right represent samples of 20 d−176 d. The symbols (*) in the figure represent significance.

**Figure 2 ijms-24-01573-f002:**
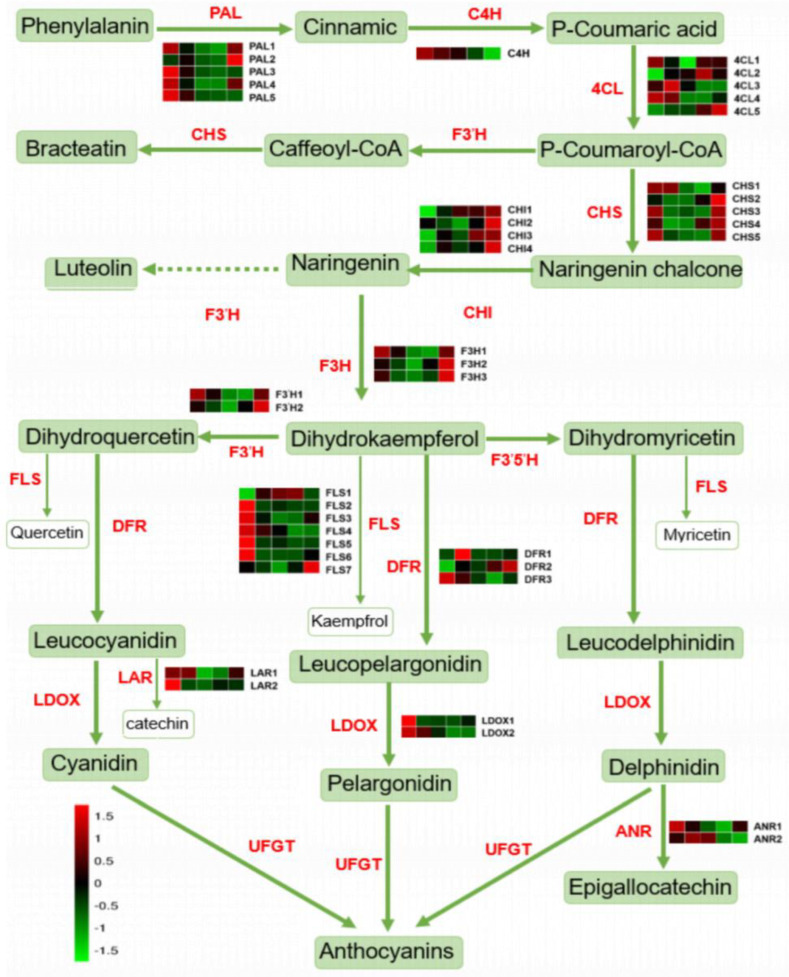
Comprehensive metabolic map of flavonoids. The gene name is on the right side of the heat map, and the FPKM value of each gene is expressed in different colors, with five cells from left to right representing samples of 20 d∓176 d, respectively; the color scale from dark blue to bright red represents relative expression levels −1.5–1.5. The name of the enzyme gene is on the side, and the FPKM of each gene is expressed in different colors and the five cells from left to right represent samples of 20 d–176 d.

**Figure 3 ijms-24-01573-f003:**
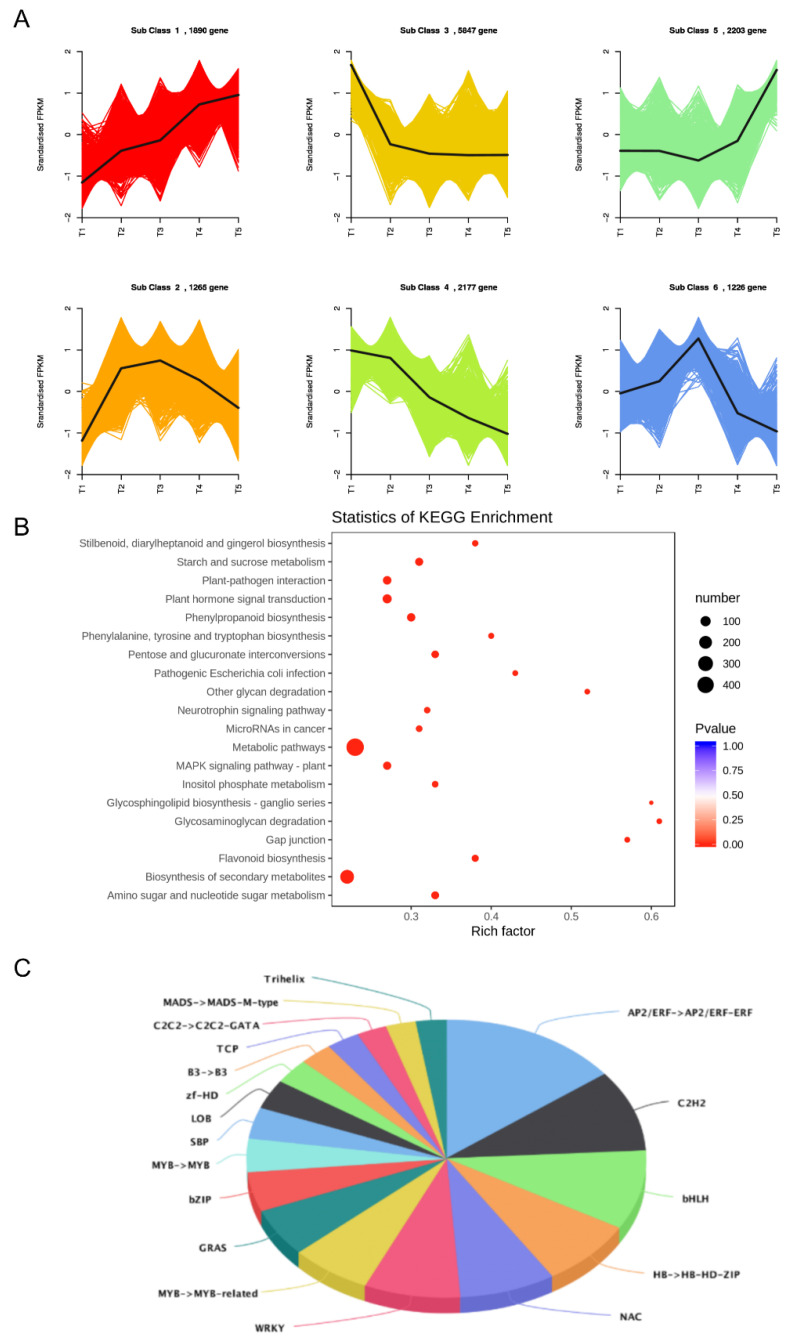
(**A**): six differential gene expression classification. The abscissa represents the sample, and the ordinate represents the standardized expression amount. (**B**): KEGG enrichment analysis of Class 3 gene. (**C**): Transcription factor families enriched in the cluster.

**Figure 4 ijms-24-01573-f004:**
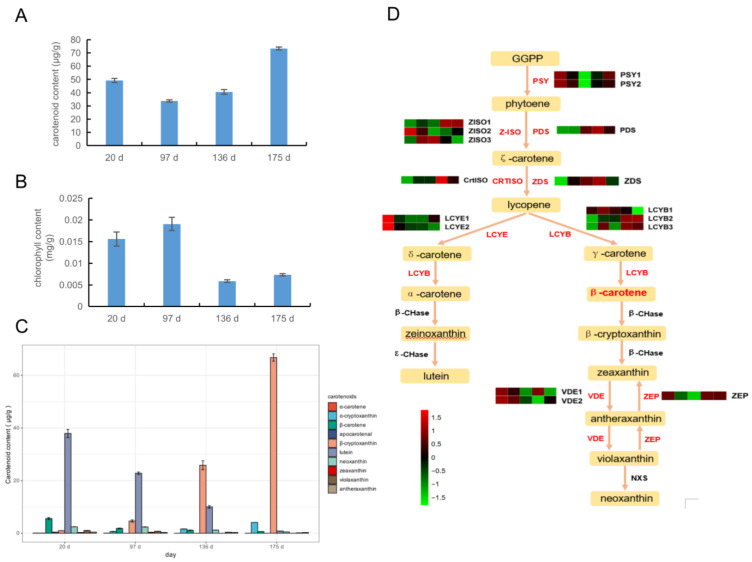
Comprehensive analysis of carotenoids. (**A**): Carotenoids content in four periods. The X axis is the content of carotenoids (μg/g), and the Y axis is the carotenoids. (**B**): Chlorophyll content in four periods. The X axis is four periods, and the Y axis is chlorophyll content (mg/g). The error bars in the figure is standard error. (**C**): Various carotenoid contents in four periods. The X axis is the content of carotenoids (μg/g), and the Y axis is the type of carotenoids. (**D**): Comprehensive metabolic map of carotenoids. The gene name is on the right side of the heat map, and the FPKM value of each gene is expressed in different colors, with five cells from left to right representing samples of 20 d–176 d, respectively, and the color scale from dark blue to bright red represents relative expression levels −1.5–1.5. The name of the enzyme gene is on the side, and the FPKM of each gene is expressed in different colors; the five cells from left to right represent samples of 20 d–176 d.

**Figure 5 ijms-24-01573-f005:**
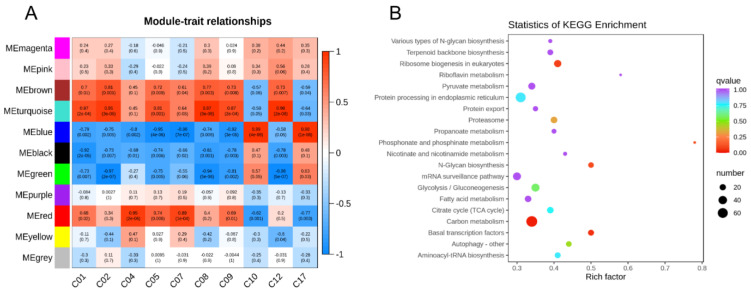
Screening for carotenoid transcription factors. (**A**): Module–flavonoid relationship. Each row represents a module, and the number of genes in each module is shown on the left. Each column represents a specific carotenoid. The value in each cell at the row-column intersection represents the correlation coefficient between the module and the flavonoid and is displayed according to the color scale on the right. The value in parentheses in each cell represents the *p* value. (**B**): KEGG enrichment analysis of “blue” module gene.

## Data Availability

The dataset generated during and/or analyzed during the current study are available from the corresponding author on reasonable request.

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
