# Peer review of "Integrative Analysis of Metabolome and Transcriptome Reveals the Mechanism of Color Formation in Yellow-Fleshed Kiwifruit"

_ijms, 2023, doi:10.3390/ijms24021573_

Round 1
Reviewer 1 Report (Previous Reviewer 2)
I found the manuscript represented a highly technical paper that did not bring out the importance and value of the research. There needs to be reflection on the value of the components at different maturation stages especially the antioxidant effects. Also colour is only one factor relating to consumer acceptance, some some indication of the texture of the fruit at the different maturation stages.
If these a points are addressed, I am otherwise satisfied with the manuscript
Author Response
Response to Reviewers We would like to thank you for your careful reading, helpful comments, and constructive suggestions, which has significantly improved the presentation of our manuscript. The following information is modified. We have carefully considered all comments from the reviewers and revised our manuscript accordingly. The manuscript has also been double-checked, and the typos and grammar errors we found have been corrected. In the following section, we summarize our responses to each comment from the reviewers. We believe that our responses have well addressed all concerns from the reviewers. We hope our revised manuscript can be accepted for publication. Question 1 I found the manuscript represented a highly technical paper that did not bring out the importance and value of the research. There needs to be reflection on the value of the components at different maturation stages especially the antioxidant effects. Also colour is only one factor relating to consumer acceptance, some some indication of the texture of the fruit at the different maturation stages. If these a points are addressed, I am otherwise satisfied with the manuscript. Response1 Thank you very much for your review comments. Your suggestions have expanded the value of our research. Indeed, color reflects the texture of fruit, especially in relation to its antioxidant properties. In our study, we focused on the color display of kiwifruit in two periods. 1. Which important anthocyanins and various flavonoid metabolites affect the formation and disappearance of pink. 2. How various carotenoids accumulate in fruits and affect the formation of fruit color. Anthocyanins and carotenoids both have antioxidant effects. Therefore, we expanded a small part of the introduction on the antioxidant of anthocyanins and carotenoids, which provided a reference for the later study of the pigment antioxidant of kiwifruit. Thanks again for your suggestion, which enriches our research content.

Reviewer 2 Report (New Reviewer)
The manuscript “Integrative analysis of metabolome and transcriptome reveals the mechanism of color formation in yellow-fleshed kiwifruit” identified that the color of yellow-fleshed kiwifruit in different periods is controlled by the metabolism of flavonoids, carotenoids, and chlorophyll.
The experiments were well done but I had some concerns regarding data representation and clarity.
1. Figure 1 only has 2 panels in the figure whereas the legend has Fig 1B and 1C in the description. I can’t seem to locate Fig. 1C in the diagram. Please include that. Regarding the heat map (Fig.1B), the legend doesn’t specifically say what the flow chart on the left of the heat map represents. Also, I understand the significance of writing the groups at the bottom of the heat map (T1-T5); however, I don’t think the color coding of the groups is necessary. It is confusing to relate the group colors with the heat map. My suggestion is to remove the group color codes.
2. Lines 148-150 are highlighted in yellow in the manuscript. Is there a specific reason for that? If not, the highlight should be removed.
3. It would be good to include more recent references. I could only see 4 references from the last 5 years.
4. Figure 2 legend looks repetitive. It is not clear what you are trying to say. My suggestion is to modify the legend in a simpler way.
5. Figure 4: Standard deviations in Fig. 4A and 4B look quite similar. I would suggest authors to check it once. I would also suggest labelingss the X-axis with units.
Author Response
Response to Reviewers
We would like to thank you for your careful reading, helpful comments, and constructive suggestions, which has significantly improved the presentation of our manuscript. We are still lacking in the presentation of data. Therefore, we have checked and modified the data in the figure according to your suggestions. In addition, we have also realized that the graphics display is not concise and neat enough. Therefore, in addition to the details you raised, we also retypeset the graphics to make the display more appropriate. We are still in the process of learning how to interpret and present the data. Your suggestions are very impressive to us and have greatly improved us. The following is our reply to your review comments.
Question 1
Figure 1 only has 2 panels in the figure whereas the legend has Fig 1B and 1C in the description. I can’t seem to locate Fig. 1C in the diagram. Please include that. Regarding the heat map (Fig.1B), the legend doesn’t specifically say what the flow chart on the left of the heat map represents. Also, I understand the significance of writing the groups at the bottom of the heat map (T1-T5); however, I don’t think the color coding of the groups is necessary. It is confusing to relate the group colors with the heat map. My suggestion is to remove the group color codes.
Response1
Thank you very much for your careful advice. We've removed the colors from the groups, so it's really clearer. The errors in the title of the figure were also corrected.
Question 2
Lines 148-150 are highlighted in yellow in the manuscript. Is there a specific reason for that? If not, the highlight should be removed.
Response2
This is highlighted due to the journal's requirement to highlight the selection of the detection period based on a paper we published previously.
Question 3
It would be good to include more recent references. I could only see 4 references from the last 5 years.
Response 3
Thank you for your advice, we refer to the necessary and recent references.
Question 4
Figure 2 legend looks repetitive. It is not clear what you are trying to say. My suggestion is to modify the legend in a simpler way.
Response 4
This figure is the main metabolic pathway of flavonoids. The heatmap in the pathway represents the relative expression level of genes, and the legend is intended to illustrate the level of gene expression.
Question 5
Figure 4: Standard deviations in Fig. 4A and 4B look quite similar. I would suggest authors to check it once. I would also suggest labelingss the X-axis with units.
Response 5
Thank you very much for your detailed advice. To avoid this error, we checked the original data, recalibrated the errors in the images, and marked the X-axis in units, but they still looked similar.

Reviewer 3 Report (New Reviewer)
Abstract
Line 11: Provide the scientific name for yellow fleshed kiwifruit right after its first mention
Line 12-13: add “the” before “metabolic level”, combine metabolic and transcription and remove “level” mentioned twice
Line 14: “development stages” and time points (20 d, 58 d etc.) do not go hand in hand. These are time points, not development stages
Line 14: What does “flavonoid metabolomics and transcriptome detection” imply? Should that be changed to flavonoid detection at metabolome and transcriptome levels?
Line 20: Change “not only due” to “in addition to”
Line 20-21: Replace “but linear carotenoids were not to” with “with none of them being linear carotenoids”
Line 21: Add the term “process” after “development”
Line 35-37: The sentence starting with “To explore….color change” does not seem complete. Similarly, the sentence between lines 44-46
Line 55: Add a reference for “The carotenoid content of A. macrosperma and its hybrids with A. melanandra is higher at 20 d” and change “hybrids” to “hybrid”
Line 56: “Increase, and its content directly affects the flesh color”. Is this a complete sentence?
Line 57: “β-carotene content increases rapidly during fruit ripening and ripening”, one “ripening” should be removed?
Line 58: Sentence formation is not correct for “Two molecules of geranyl geranyl pyrophosphate (GGPP) to form phytoene, catalysed by the enzyme phytoene synthase (PSY).”
Materials and Methods:
Line 81: What kind of extract was used to dissolve the powder?
Line 81: Change “is stored” to “was stored”
Line 85: First sentence is incomplete
Line 93: Re-write “Repeat the above steps after collecting the supernatant” with correct verbal forms
Line 111: Add a reference for the CTAB method
Line 112: Which Qubit kit was used, BR or HS
For RNA isolation, how many samples and replicates were used? Which part of the fruit and how much of it was used for each isolation?
Add the company location for Illumina Hiseq platform
Line 128: Re-write “Use the WGCNA (weighted gene co-expression network analysis) package in the R package to construct a gene co-expression network”. Same thing for the sentence between lines 131-132 and line 133-134
Line 137: Spell out “DAA”
Line 138: “The first strand of cDNA was synthesized using reverse transcriptase” does not seem complete
Reaction conditions for qRT-PCR have not been provided
Line 146: Change “fruit were” to “fruit was”
Results:
The first part of transcriptome profiling results does not seem to be related where the authors have talked about phenotypic characteristics and can be moved to another section.
What was the control group for transcriptome analysis? T1, 20d? or treatments were compared to each other?
Discussion:
Line 233-252: Seems like introduction and should be removed from the discussion section. The discussion is about re-iterating the major findings of the current study and comparing those with other studies (currently lacking)
Author Response
Response to Reviewers
We would like to thank you for your careful reading, helpful comments, and constructive suggestions, which has significantly improved the presentation of our manuscript. The following information is modified.
Question 1(abstract)
Line 11: Provide the scientific name for yellow fleshed kiwifruit right after its first mention.
Line 12-13: add “the” before “metabolic level”, combine metabolic and transcription and remove “level” mentioned twice.
Line 14: “development stages” and time points (20 d, 58 d etc.) do not go hand in hand. These are time points, not development stages.
Line 14: What does “flavonoid metabolomics and transcriptome detection” imply? Should that be changed to flavonoid detection at metabolome and transcriptome levels?
Line 20: Change “not only due” to “in addition to”.
Line 20-21: Replace “but linear carotenoids were not to” with “with none of them being linear carotenoids”.
Line 21: Add the term “process” after “development”.
Response 1
Thank you very much for your careful reading. The questions in the abstract have been revised and the changes have been highlighted.
Question 2(introduction part)
Line 35-37: The sentence starting with “To explore….color change” does not seem complete. Similarly, the sentence between lines 44-46.
Line 55: Add a reference for “The carotenoid content of A. macrosperma and its hybrids with A. melanandra is higher at 20 d” and change “hybrids” to “hybrid”.
Line 56: “Increase, and its content directly affects the flesh color”. Is this a complete sentence?
Line 57: “β-carotene content increases rapidly during fruit ripening and ripening”, one “ripening” should be removed?
Line 58: Sentence formation is not correct for “Two molecules of geranyl geranyl pyrophosphate (GGPP) to form phytoene, catalysed by the enzyme phytoene synthase (PSY).”
Response 2
Thank you very much for your careful reading. For unclear and incomplete sentences, we have modified them. The questions in the introduction have been revised and the changes have been highlighted.
Question 3 (materials and methods part)
Line 81: What kind of extract was used to dissolve the powder?
Line 81: Change “is stored” to “was stored”.
Line 85: First sentence is incomplete.
Line 93: Re-write “Repeat the above steps after collecting the supernatant” with correct verbal forms.
Line 111: Add a reference for the CTAB method.
Line 112: Which Qubit kit was used, BR or HS? For RNA isolation, how many samples and replicates were used? Which part of the fruit and how much of it was used for each isolation?Add the company location for Illumina Hiseq platform.
Line 128: Re-write “Use the WGCNA (weighted gene co-expression network analysis) package in the R package to construct a gene co-expression network”. Same thing for the sentence between lines 131-132 and line 133-134.
Line 137: Spell out “DAA”.
Line 138: “The first strand of cDNA was synthesized using reverse transcriptase” does not seem complete. Reaction conditions for qRT-PCR have not been provided.
Response 3
Thank you very much for your careful reading and pertinent suggestions. In the methods section, we have corrected the experimental description and the changes have been highlighted. Total RNA was extracted from flesh tissues of the outer pericarp. RNA concentration and purity were determined by a Qubit® RNA Assay Kit (BR). For each 100 mg of tissue, 1 mL of chilled Trizol was added. The RNAseq experiment included three biological replicates per treatment.
Question 4 (results part)
Line 146: Change “fruit were” to “fruit was”.
The first part of transcriptome profiling results does not seem to be related where the authors have talked about phenotypic characteristics and can be moved to another section.What was the control group for transcriptome analysis? T1, 20d? or treatments were compared to each other?
Response 4
“fruit were” has been corrected.
Thank you for your careful reading and thinking. The control group in the experiment was T1. This paper focuses on finding the causes of color change first at the metabolic level and then at the genetic level. Both flavonoid metabolites and carotenoid metabolites were combined with the transcriptome. In order to correlate with subsequent results, we briefly described the results of transcriptome determination in order to show that the sequencing results were qualified. This result also describes the state of the fruit, which is related to the phenotype.
Question 5 (discussion part)
Line 233-252: Seems like introduction and should be removed from the discussion section. The discussion is about re-iterating the major findings of the current study and comparing those with other studies (currently lacking).
Response 5
Thank you very much for your suggestion. We have deleted the description at the beginning of the discussion section and added the description about ‘Jinshi 1’ in the background section.In the discussion section, we compared the research results of others to get our own predictions and conclusions, which we highlighted.In addition, we reviewed newer literature with higher research relevance and added new literature to supplement the current research.
The following is a brief description of our research results, which is also the highlight of our research. Perhaps there are still some deficiencies. We will try our best to extract effective information based on the existing experimental results.
1.76% of flavonoids were synthesized and accumulated in large quantities at 20 day after full bloom.
2.The red flesh at the young fruit stage is mainly caused by the accumulation of pelargonidin and cyanidin and their derived anthocyanins.
3.The increase in carotenoids and the decrease in chlorophyll together cause kiwifruit to change from green to yellow.
4.Ten carotenoids were identified, β-cryptoxanthin is the most abundant carotenoid in kiwifruit.

Round 2
Reviewer 3 Report (New Reviewer)
Abstract
Line 12: change “analysis” to “analyze”
Line 22: change “during the 175 d” to “at day 175”
Line 22-23: change numerical “7” to “10” to words
Materials and Methods
Line 96: “and the last supernatant was collected” what does this mean?
What could be a follow up study? The abstract provides a brief mention of how the study serves as the data reference for the detection of metabolites in the kiwifruit flesh. Explain a bit more on this in the conclusion section.
Author Response
Response to Reviewers
We would like to thank you for your careful reading, helpful comments, and constructive suggestions, which has significantly improved the presentation of our manuscript. The following is our reply to your review comments.
Question 1
Line 12: change “analysis” to “analyze”
Line 22: change “during the 175 d” to “at day 175”
Line 22-23: change numerical “7” to “10” to words
Line 96: “and the last supernatant was collected” what does this mean?
Response1
Thank you very much for your careful advice. The errors were also corrected.
Question 2
What could be a follow up study? The abstract provides a brief mention of how the study serves as the data reference for the detection of metabolites in the kiwifruit flesh. Explain a bit more on this in the conclusion section.
Response2
Thank you very much for your thoughtful advice.I understand that the conclusion of the abstract needs to be more detailed, not the conclusion of the article. So I added a little bit of content to the conclusion of the abstract.
This study was the first to analyze the effect of flavonoid accumulation on the pink color of yellow-fleshed kiwifruit. The high proportion of β-cryptoxanthin in yellow-fleshed kiwifruit was preliminarily found. This provides information on metabolite accumulation for further revealing the pink color of yellow-fleshed kiwifruit, and also provides a new direction for the study of carotenoid biosynthesis and regulation in yellow-fleshed kiwifruit.

This manuscript is a resubmission of an earlier submission. The following is a list of the peer review reports and author responses from that submission.
Round 1
Reviewer 1 Report
The authors performed metabolome and transcriptome analysis of kiwifruit flesh of the variety Junshi 1. The manuscript documented the observed changes of metabolome and transcriptome during fruit ripening, it is not seemed to answer the scientific questions raised in introduction section.
Author Response
Response to Reviewer 1
We would like to thank you for your careful reading, helpful comments, and constructive suggestions, which have significantly improved the presentation of our manuscript. We have carefully considered all the comments from the reviewers and have revised our manuscript accordingly. The manuscript has also been double-checked, and any grammar errors we found have been corrected. In the following section, we summarize our responses to each comment from the reviewers. We hope our revised manuscript can be accepted for publication.
Question 1
The authors performed metabolome and transcriptome analysis of kiwifruit flesh of the variety Junshi 1. The manuscript documented the observed changes of metabolome and transcriptome during fruit ripening, it is not seemed to answer the scientific questions raised in introduction section.
Response1
Thank you very much for your review. We have made a lot of revisions to the manuscript. The results and the structure of the discussion part and the way of discussion were readjured. The focus of our study was to analyze the influence of the increase and decrease of metabolites on the color change of yellow meat kiwifruit, and then to screen the transcription factors related to the synthesis of flavonoids and carotenoids.
We get the following results:
- Through the analysis of the content changes of flavonoid metabolites, it was found that the accumulation of pelargonidin and cyanidin and their respective anthocyanin derivatives was related to the pink flesh of young fruit, but not to delphinidin and its derivative anthocyanins.
- A total of 140 flavonoid compounds were detected in the flesh, among which anthocyanin and 76 % flavonoid compounds had the highest content at 20 d, and began to decrease significantly at 58 d until 175 d, resulting in the pale pink fading of the flesh. At the mature stage of fruit development (175 d), the degradation of chlorophyll and the increase of carotenoids jointly led to the change of flesh color from green to yellow, not only due to chlorophyll degradation.
- In kiwifruit flesh, 10 carotenoids were detected, but linear carotenoids were not. During the whole development of kiwifruit, the content of β-carotene was always higher than that of α-carotene. In addition, β-cryptoxanthin was the most accumulated pigment in the kiwifruit during the picking period (175 d).
- Through transcriptome analysis of kiwifruit flesh, 7 key transcription factors for flavonoid biosynthesis and 10 key transcription factors for carotenoid synthesis were screened. In short, the color of yellow fleshed kiwifruit in different periods is controlled by the metabolism of flavonoids, carotenoids and chlorophyll.

Reviewer 2 Report
I found this an interesting project.
Colour is obviously important for consumer purposes but what about flavour? I feel some reference needs to be made yo flavour eg no difference, some difference or major difference.
My general comment is that the syntax needs to be checked.
Author Response
Response to Reviewer 2
We would like to thank you for your careful reading, helpful comments, and constructive suggestions, which have significantly improved the presentation of our manuscript. We have carefully considered all the comments from the reviewers and have revised our manuscript accordingly. The manuscript has also been double-checked, and any grammar errors we found have been corrected. In the following section, we summarize our responses to each comment from the reviewers. We hope our revised manuscript can be accepted for publication.
Question 1
Colour is obviously important for consumer purposes but what about flavour? I feel some reference needs to be made yo flavour eg no difference, some difference or major difference.
My general comment is that the syntax needs to be checked..
Response1
Thank you very much for your review, including the topic of flavor, which is very meaningful to us. In this manuscript, we mainly address the scientific question of what compounds in yellow meat kiwifruit determine its flesh color, and screen out some key transcription factors. Of course, both color and flavor are important commercial values of kiwifruit, and we will present a richer flavor study in the next manuscript.
We get the following results:
- Through the analysis of the content changes of flavonoid metabolites, it was found that the accumulation of pelargonidin and cyanidin and their respective anthocyanin derivatives was related to the pink flesh of young fruit, but not to delphinidin and its derivative anthocyanins.
- A total of 140 flavonoid compounds were detected in the flesh, among which anthocyanin and 76 % flavonoid compounds had the highest content at 20 d, and began to decrease significantly at 58 d until 175 d, resulting in the pale pink fading of the flesh. At the mature stage of fruit development (175 d), the degradation of chlorophyll and the increase of carotenoids jointly led to the change of flesh color from green to yellow, not only due to chlorophyll degradation.
- In kiwifruit flesh, 10 carotenoids were detected, but linear carotenoids were not. During the whole development of kiwifruit, the content of β-carotene was always higher than that of α-carotene. In addition, β-cryptoxanthin was the most accumulated pigment in the kiwifruit during the picking period (175 d).
- Through transcriptome analysis of kiwifruit flesh, 7 key transcription factors for flavonoid biosynthesis and 10 key transcription factors for carotenoid synthesis were screened. In short, the color of yellow fleshed kiwifruit in different periods is controlled by the metabolism of flavonoids, carotenoids and chlorophyll.

Round 2
Reviewer 1 Report
Dear authors,
Figures 1A and 1B are already presented at https://doi.org/10.1002/jsfa.10251. This is a serious problem called self-plagiarism.